# A New Perspective on Huntington’s Disease: How a Neurological Disorder Influences the Peripheral Tissues

**DOI:** 10.3390/ijms23116089

**Published:** 2022-05-29

**Authors:** Laura Gómez-Jaramillo, Fátima Cano-Cano, María del Carmen González-Montelongo, Antonio Campos-Caro, Manuel Aguilar-Diosdado, Ana I. Arroba

**Affiliations:** 1Undad de Investigación, Instituto de Investigación e Innovación en Ciencias Biomédicas de la Provincia de Cádiz (INiBICA), 11002 Cádiz, Spain; laura.gomez@inibica.es (L.G.-J.); fatima.cano@inibica.es (F.C.-C.); mcarmen.gonzalez@inibica.es (M.d.C.G.-M.); antonio.campos@uca.es (A.C.-C.); manuel.aguilar.sspa@juntadeandalucia.es (M.A.-D.); 2Área de Genética, Departamento de Biomedicina, Biotecnología y Salud Pública, Universidad de Cádiz, 11002 Cádiz, Spain; 3Departamento de Endocrinología y Nutrición, Hospital Universitario Puerta del Mar, Universidad de Cádiz, 11002 Cádiz, Spain

**Keywords:** Huntington’s disease, peripheral tissues, animal models, experimental approaches

## Abstract

Huntington’s disease (HD) is a neurodegenerative disorder caused by a toxic, aggregation-prone expansion of CAG repeats in the HTT gene with an age-dependent progression that leads to behavioral, cognitive and motor symptoms. Principally affecting the frontal cortex and the striatum, mHTT disrupts many cellular functions. In fact, increasing evidence shows that peripheral tissues are affected by neurodegenerative diseases. It establishes an active crosstalk between peripheral tissues and the brain in different neurodegenerative diseases. This review focuses on the current knowledge of peripheral tissue effects in HD animal and cell experimental models and identifies biomarkers and mechanisms involved or affected in the progression of the disease as new therapeutic or early diagnostic options. The particular changes in serum/plasma, blood cells such as lymphocytes, immune blood cells, the pancreas, the heart, the retina, the liver, the kidney and pericytes as a part of the blood–brain barrier are described. It is important to note that several changes in different mouse models of HD present differences between them and between the different ages analyzed. The understanding of the impact of peripheral organ inflammation in HD may open new avenues for the development of novel therapeutic targets.

## 1. Introduction

Increasing evidence shows that peripheral tissues are affected by neurodegenerative diseases [1]. The research on neurodegenerative disease progression has been focused on the mechanisms that occur in neurons and nearby support cells in the brain. Nowadays, it has been established that there is active crosstalk between peripheral tissues and the brain in different neurodegenerative diseases, such as Parkinson’s disease [2], Alzheimer’s disease [3] and prion diseases [4].

Huntington’s disease (HD) is a devastating neurodegenerative disorder caused by a toxic, aggregation-prone expansion of the trinucleotides (cytosine/adenosine/guanine (CAG)) encoding for glutamine (Q) residues in the huntingtin (HTT) gene [5], with an age-dependent progression that leads to behavioral, cognitive and motor symptoms. Whereas normal, non-pathogenic HTT has between 6 and 35 glutamine residues, mutant HTT (mHTT) has >36–250. Similar to wild-type HTT, mutant HTT is expressed in most tissues [6].

In HD patients, gamma-amino butyric acid (GABA)-releasing medium spiny neurons that are primarily expressed in the D2 dopamine receptor demonstrate an enhanced susceptibility to neurodegeneration, leading to the development of chorea [7]. Apart from the neuronal impairment of HD, some studies have described pathological phenotypes in the peripheral tissues of HD patients, including altered glucose homeostasis [8] and weight loss [9]. Other studies have analyzed the particular changes in blood cells such as erythrocytes [10] and lymphocytes [11], as well as in fibroblasts [12], immune blood cells [13,14], the pancreas [15], the heart [16,17], the retina [18], the liver [19], the kidney [20], pericytes and the blood–brain barrier [21] from HD patients. The models can aid understanding of how HTT affects peripheral tissues and contributes to disease progression. The potential peripheral effects have escaped notice due to the strong neural effects in HD patients. The discovery that normal and mHTT are expressed in many tissues and organs in humans and other mammals has given rise to the hypothesis that HD could be studied as a neural disease with potent and unknown systemic effects, rather than only as a condition related to central nervous system (CNS) dysfunctions [22]. 

In the study of HD over the years, several animal models have been developed to recapitulate the genetic and phenotypic aspects of the disease [23,24]. Experimental models have been studied using yeast, *Caenorhabditis elegans*, Drosophila [25] and mouse genetic models expressing the mutant HTT gene [26]. However, not only can animal models provide a way to understand the pathology of the disease, but different specific tissue cultures are also included in the most influential experimental approaches.

Actually, there are several mouse models which are generated by randomly inserting into its genome exon 1 of human HTT with 116 and 144 CAG repeats, respectively, that are ubiquitously expressed under the human HTT promoter as R6/1 and R6/2 [26]. The N-171-82Q model contains a longer fragment of the protein HTT than the one carried by the R6, carrying exons 1 and 2 (the first 171 amino acids) with 82 polyQ repeats under the control of the protein promoter of the murine prion, so expression occurs throughout the brain but is restricted to neurons [27]. The HD transgenic models 19Q and 84Q include exon 1 of the human HTT gene with the expansion of 84 and 19 CAG trinucleotide repeats, respectively, inside GFP-HTT fusion genes under the control of a human ubiquitin promoter [28]. Additionally, BACH mice present 97 CAA-CAG repeats [29]. Other mouse models involve mice into which the complete HTT protein has been inserted and present two copies of normal murine HTT and one copy of HTT, such as the YAC mouse model. The YAC128 model contains pathogenic expansions of 128 CAG repeats [30], and the zQ175 mouse presents 175 CAG repeats [31] under the control of the promoter of human HTT [32].

The knock-in genetic models represent human HD more faithfully because they are obtained by introducing the mutation causing HD (polyQ expansion) within the orthologous gene itself, murine HTT, in the exact position it would occupy if the mutation occurred naturally. The result is that these mice have two copies of the HTT protein, with either both modified (homozygous) or only one modified (heterozygotes), with both under the direction of the endogenous promoter of HTT. There are several types of knock-in models with expansions of polyQ, ranging from 50 [33] to 150 repeats [34]. The HdhQ150 model was generated by knocking in an expanded repeat of 150 CAGs into the mouse HTT gene [34]. The CAG140 knock-in mouse has 140 CAG repeats [26].

This review focuses on current knowledge of peripheral tissue effects in HD animal and cell experimental models and identifies biomarkers and mechanisms involved or affected in the progression of the disease as new therapeutic or early diagnostic options. It is important to note that several changes in different mouse models of HD present differences between them and between the different ages analyzed. Understanding the impact of peripheral organ inflammation in HD may open new avenues for the development of novel therapeutic targets. 

## 2. Serum/Plasma

It is well established that changes in circulating levels of cytokines are closely correlated with HD progression [35]. Different studies support the importance of blood markers and their relationship with the inflammatory response in peripheral organs. Murine models of HD are powerful tools for studying potential molecular and cellular mechanisms underlying HD pathogenesis [36]. However, the different animal models analyzed showed differential patterns of cytokines along with HD progression. 

BACHD mice (12-month-old) presented no changes in their serum levels of cytokines [37], and the quantification of protein levels in blood from 8-week-old R6/2 mice showed no increase in cytokine levels; however, at 14 weeks old, the IL-6, IL-8, IL-1β and TNF-α levels in R6/2 mice were elevated in the same way as anti-inflammatory cytokines IL-2 and IL-10 [38], which correlates with HD disease severity [39]. A study including serum samples of YAC128 mice reveals significant increases in interleukin-6 (IL-6), CXCL1 (a functional homolog of IL-8 in mice), interferon-γ (IFN-γ) and IL-10 [13]. BAC-225Q mouse plasma between 2 and 10 months of age showed increased levels of IL-8, but for IFNγ, IL-6 and TNFα, no genotype-dependent alterations were found to be associated [40]. The elevated serum cytokine levels are associated with the late stage and severity of the disease in the animal model analyzed.

## 3. Innate and Adaptive Immune System in HD

Mutant HTT (mHTT) induces immune activation in a ubiquitous way in the CNS and the peripheral immune system (blood cells) and at the molecular level (intracellular signaling). At present, the contribution of the immune system to the pathology of HD has gained interest due to the development of immune-based therapy tools [38]. HTT is more highly expressed in neurons as compared to other cell types. However, immune cells, including B cells, T cells, monocytes and/or macrophages, from patients show detectable levels of HTT that are positively correlated with disease status [38]. Immune cells such as monocytes produce inflammatory cytokines, and they can differentiate into pro-inflammatory dendritic cells (DCs) or macrophages [41].

Usually, the innate immune system plays a beneficial role, although sometimes this system has a negative impact on the organism by promoting an uncontrolled increase in inflammation [42]. In this context, a direct toxic effect promoted by mHTT induces an inflammatory response, and both the innate and adaptive immune systems could play an important role in HD. The possibility of using immune cells from HD patients with different progression statuses of the disease as an ex vivo experimental approach presents the opportunity to understand the impact of HD on peripheral immune tissues.

### 3.1. Macrophages

The regulation of inflammation and the innate immune response is controlled by macrophages as critical effectors. Macrophages are pleiotropic immune cells with roles in homeostasis and inflammation, such as phagocytosis and the production of cytokines and other inflammatory mediators [43]. Macrophages are able to secrete pro- and anti-inflammatory cytokines and chemokines, which communicate signals to surrounding cells. In general, the concept of M1/M2 macrophages postulates that M1 macrophages are assumed to have pro-inflammatory functions, and M2 macrophages have anti-inflammatory and wound-healing functions. Polarization between M1/M2 phenotypes has a critical therapeutic value, especially in diseases in which M1/M2 imbalance plays a pathophysiological role [44]. Thus, further studies on M1/M2 macrophages in HD are of great interest, as is the understanding of HTT levels and macrophage function [45]. 

In a Drosophila model, the expression of mHTT in hemocytes altered the immune response by causing deficient cellular and humoral immune responses against pathogens. The presence of mHTT in Drosophila macrophage-like S2 cells in vitro reduced energy potential, phagocytic activity and the production of antimicrobial peptides, basically due to the altered production of cytokines and the activation of JAK/STAT signaling [46] (Table 1). However, macrophages from an R6/2 mouse model of HD (14-week-old) presented elevated IL-4, IL-10, IL1-2 and IL-6 levels compared to wild-type (WT) mice, showing the potentiation of the M2-like phenotype [38]. The expression of activation markers for macrophages was also upregulated in zQ175 mice (24-month-old) compared to WT mice. On the other hand, monocytes were different from macrophage cells in HD patients and HD mouse models; when cultured with a pro-inflammatory stimulus, they secreted higher IL-6 levels compared to control or WT macrophages. These results showed that mHTT in monocyte/macrophage cells induces a hyperactive response under deleterious stimulation [39] (Table 1).

Bone marrow transplants as a source of progenitors of peripheral blood monocytes in HD mouse models such as YAC128 and BACHD mice replace the peripheral immune system. A transplanted HD mouse model attenuated kinetic symptoms, and the cytokine levels were similar to those of WT mice [47]. In R6/1 mouse models, studies have been performed using bone-marrow-derived mast cells (BMMC). Mast cells are derived from the yolk sac [48], and in adults, they are generated in bone marrow and migrate as immature precursors. The BMMC response in R6/1 mice under a pro-inflammatory stimulus is low compared to that in WT mice.

These results suggest that macrophage cells could be an important tool for new immunotherapeutic approaches to HD.

### 3.2. T Cells

In different neurodegenerative diseases, the role of T cells has been analyzed via the measurement of higher T-cell responses and a shift in CD4^+^ and CD8^+^ cell populations [49]. However, the involvement of T cells in HD is still unclear. T lymphocytes are a major source of cytokines, and T lymphocytes are subdivided into Th1 and Th2; the cytokines they produce are known as Th1- and Th2-type cytokines, respectively. Th1-type cytokines tend to produce the pro-inflammatory responses responsible for killing intracellular parasites and for perpetuating autoimmune responses. Interferon gamma is the main Th1 cytokine. Excessive pro-inflammatory responses can lead to uncontrolled tissue damage, so there must be a mechanism to counteract this. Th2-type cytokines include interleukins 4, 5 and 13; they are associated with the promotion of interleukin-10, which produces an anti-inflammatory response [50].

In HD patients, IL-4 levels were higher compared to healthy subjects. IL-4 is related to promoting M2 in macrophage cells or Th2 in T helper cells [14]. The increase in IL-4 detected in a mouse model at late stages of HD [39] could establish a relation between the adaptive response in chronic immune activation and the participation of critical players: T cells and monocytes/macrophages. The determination of T-cell levels in different animal models showed an increase in R6/2 and zQ175 mice at a late stage of HD compared to WT mice [51]. In order to determine the specific role of T cells during HD progression, some works have demonstrated that immunized mutant mice (R6/1 and zQ175) also exhibit the downregulation of genes that function in Th2 responses and memory T cell responses [52] at the critical early stage of the disease (Table 1).

**Table 1 ijms-23-06089-t001:** Immune cell phenotype. ⇓ Reduced levels. ⇑ Augmented levels. NA, no alterations. -, no data. a, late stage. b, early stage.

Immune Cells	Observations	Drosophila	R6/1	R6/2	zQ175	Yac128	BACHD
**Macrophages**	M1/M2 response	⇑ M2 a [46]	⇓ M1 a [48]	M2 ⇑ a [38]	⇑ M1 a [39]	⇑ M1 a [47]	⇑ M1 a [47]
**T Cells**	Th1/Th2 response	-	-	⇑ Th1 a [14]	⇑ Th1 a [14]	-	-
		-	-	⇓ Th2 b [52]	⇓ Th2 b [52]	-	-
**Dendritic Cells**	Cell number variations	-	-	⇑ cell number b [51]	⇑ cell number a [51]	-	-

### 3.3. Dendritic Cells

Dendritic cells (DCs) are antigen-processing and presenting migrating cells that regulate T-cell responses [53]. DCs present two states: an immature state with strong phagocytic activity and a mature state with high cytokine production [41]. DCs are able to migrate; however, they only have short lives [54]. In an animal model of HD, DCs contained mHTT and further fostered pathogenic immune responses [55]. DCs in the spleen or peritoneal cavity were also upregulated in late-stage zQ175 mice as compared to WT mice, as occurred in R6/2 at 8 weeks old, where an increased percentage of activated DCs was detected during the early stage of the pathology [51]. Regarding DCs’ role in the pathology of HD, information on the specific effect of DCs during HD is still lacking (Table 1). 

## 4. Spleen as Immune Tissue

It has been reported that patients with HD exhibit peripheral immune changes, including circulating and tissue-based immune cells [12]. In R6/2 mice, activated T cells can be identified by the expression of cell surface markers, such as OX40, CD40L and CD25 [56]. In the spleen, which contains T cells and a small percentage of DCs and macrophages, the expression of OX40L and CD40 was increased in R6/2 mice from 8 weeks of age onward, and high levels of splenic *IL-12* and *CD40* were detected. The cytokine gene expression of *IL-10*, *IL-17* and *IL-12* in R6/2 was increased in splenic tissue, whereas *IL-1β*, *IL-4*, *IL-6* and TNF-α levels remained unchanged [38]. However, when individual cultures from each spleen of the R6/2 mice at 12 weeks of age were stimulated with IFNγ and LPS, changes in IL-1β levels, IL-4 and TNF-α were demonstrated, but no differences in T-cell subsets were detected in R6/2 spleens when compared with those from WT mice [51]. 

The analysis of the expression levels of cytokines in splenocytes from late-stage zQ175 mice at 24 months of age showed similar increases to those observed in late-stage R6/2 mice. However, a study of the spleens from BACHD mice (12-month-old) showed differential expression in some pro- and anti-inflammatory cytokines, with no increased detection in IL-12 expression levels but with a similar pattern of expression in the other cytokines analyzed [37] compared to R6/2 and zQ175 mice. As occurs in cytokine serum levels, there is a relation between the late stages of the disease and the levels of cytokines detected in the spleen.

## 5. Kidney

### 5.1. Kidney Organ and mHTT Accumulations

Post-mortem analyses revealed the presence of HTT aggregates in the kidneys of HD patients [21]. The different HD animal models presented specific mHTT accumulations in the kidneys. In R6/2 and 150Q mice, the inclusions were only detected in the cell nucleus, with a moderate frequency of inclusion in the tubular, interstitial and glomerular cells [57,58] (Table 2). In contrast, CAG140Q (HD knock-in) mice showed no obvious mHTT in the kidney [59] due to mHTT being more stable in the brain than in peripheral tissues. In most tissues, the presence of inclusions was correlated with a progressive decrease in the size of the respective organs [57], suggesting that abnormal cell death may occur in various tissues [47]. This was dependent on the animal model used and tissue studied because, in an HTT genetically modified mouse, an HdhQ150 knock-in mouse model (150Q) presents higher weight in the kidneys compared to WT mice [58] (Table 2).

The relation between inclusion and organ weight has been a topic in studies of mHTT presence in peripheral organs. The kidney weight in R6/2 (from 6 weeks to 12 weeks) and hQ150/Q150 (at 22 months) was lower in the HD experimental mice than in their wild-type littermates. However, in the YAC128 HD mouse model, the organ weight was increased [60]. In kidney tissues from 84Q transgenic mice, but not in samples from 19Q mice, an increased level of HTT aggregates was observed [28] (Table 2).

### 5.2. Kidney Intracellular Signaling Alteration

Several intracellular signaling pathways are involved in HD progression [20]. The possible effect of mHTT accumulation during the correct cellular function has been analyzed, mainly regarding neural systems. On the other hand, there are only a few articles related to intracellular signaling dysfunction and HD progression in kidney tissue.

In several studies, mitochondrial dysfunction during HD has been analyzed [19]. However, there are only a few articles that describe how mHTT accumulation could be responsible, at least in part, for kidney failure during HD progression. The YAC128 mice showed similar O_2_ consumption, CO_2_ release, physical activity, food consumption and fasting blood glucose levels compared to the WT mice. This metabolically normal environment suggests that mitochondrial respiratory dysfunction is not essential for HD pathogenesis in the kidneys [61] (Table 2).

**Table 2 ijms-23-06089-t002:** Kidney phenotype. ⇓ Reduced. ⇑ Augmented. NA, no alterations. -, no data. a, late stage, b, early stage.

Kidney	R6/2	HdhQ150	YAC128	CAG140Q	BACHD	CAG19Q	CAG84Q
Observations							
**mHTT inclusion**	In cell nuclei of tubular, interstitial and glomerular cells [57] b	In cell nuclei of tubular, interstitial and glomerular cells [58] a	-	No detection [59] a	-	No detection [28] a	Yes [28] a
**Organ weight**	⇓ [57] b	⇓ [58] a	⇑ [60] a	-	-	-	-
**Mitochondrial effects**	-	-	NA [61] a	-	-	-	-
**Inflammatory processes**	-	-	-	-	⇑ IL-6 [37] a	-	-
**UPS and autophagy system**	-	-	-	-	-	NA [28] a	NA [28] a

Recently, there has been increased interest in the inflammatory processes associated with Huntington’s disease [62]. The role of inflammation in kidney failure has been described in multiple neurodegenerative disorders [63,64,65,66]. The BACHD mice showed significant changes in IL-6 cytokine levels in the kidneys, in comparison with WT animals [37]. This cytokine is directly associated with the progression of chronic kidney disease [67].

Researching one possible reason for the abnormal mHTT protein aggregation in CNS and peripheral tissues, some studies have demonstrated that there is inadequate protein degradation in HD models [68]. However, impairment of the ubiquitin–proteasome system (UPS) or autophagy in peripheral tissues was not as severe as in the CNS. These results suggest that the protein clearance pathway is also disturbed in the peripheral tissues of HD models, leading to the dysfunction of different organs. Therefore, mHTT may interfere with the basic transcriptional mechanisms of many genes [47]. Supporting this theory, enhancing autophagy induces beneficial effects in cellular, fly and mouse models of HD [69,70]. However, the UPS and autophagy system in kidney tissues from 19Q and 84Q transgenic mice were not affected at different times of the study, but the presence of mHTT was only detected in 84Q transgenic mice [28]. 

In order to understand this potential therapeutic protein clearance system inside the kidney in HD models, several articles have studied human embryonic kidney cells (HEK293) expressing GFP-tagged, HTT-exon1-containing 74Q [71] or HEK293FT cells expressing 84Q or 19Q [28] as cellular models in which UPS induction might reproduce earlier in kidney tissues of HD models (Table 2).

## 6. Liver

### 6.1. Liver Organ and mHTT Accumulations

Previously, it has been noted that mHTT accumulation in peripheral organs or tissues affects the liver, among other organs [19]. The presence of mHTT was detected in the liver in different animal models, such as YAC128 [60] and CAG140Q (HD knock-in) [59]; in R6/2 and HdhQ150 knock-in mouse models of HD, it was mainly found in hepatocytes, but only in R6/2 mice was it found in the bile duct epithelium [58]. In all of these models, weight loss was not uniform across all tissues, being most evident in the liver and least evident in the kidneys in R6/2 mice [57], in contrast with 150Q mice [58]. Moreover, in liver tissues, an increased level of mHTT aggregates over time was displayed in 84Q mice; similar to other tissues, mHTT was not found in 19Q transgenic mice [28] (Table 3).

Several observations of liver morphology in HD patients revealed a minor failure or disturbance of liver function [72]. Hepatocytes from HD patients were smaller than normal and had a shortened survival time [19,73]. Regarding the differential sizes of the organs [40], the animal model analyzed defines the significant or non-significant differences between organ weights [58].

### 6.2. Liver Intracellular Signaling Alteration

The inflammatory processes may be involved in HD progression (see immune system paragraphs). To examine this possibility, the levels of inflammatory and regulatory cytokines in the livers of 12-month-old BACHD mice were enhanced (IL-12p70 and TNF-α) compared to WT mice [37].

The implication of the UPS during HD progression has been analyzed in CNS and affected peripheral organs [74]. Specifically, regarding liver tissues, the UPS system in 19Q and 84Q transgenic mice was impaired in the liver [28].

Additional whole-organism effects induced by mHTT include a decline in systemic metabolic homeostasis, which stems from the derangement of the liver. Both R6/2 and HdhQ150 knock-in mice showed deficiency in the urea cycle, with hyperammonemia, high blood citrulline levels and the suppression of urea cycle enzymes as prominent features of HD [75]. Transfections with constructs of mHTT-25Q and mHTT-109Q demonstrated that those aggregates suppressed the expression of the C/EBPa transcription factor in HepG2 cells and, moreover, recruited this factor. Consequently, C/EBPa lost its ability to interact with the CREB-binding protein cofactor, which appears to mediate urea cycle deficiencies [76]. There was an association between high circulating concentrations of ammonia and a failure in the transcription of urea-cycle enzymes in the livers of mice with HD [22] (Table 3). 

**Table 3 ijms-23-06089-t003:** Liver phenotype. ⇓ Reduced. ⇑ Augmented. NA, no alterations. Hep., Hepatocytes. -, no data. a, late stage. b, early stage.

Liver	R6/2	HdhQ150	YAC128	CAG140Q	BACHD	CAG19Q	CAG84Q	N171-82Q
Observations								
**mHH inclusion**	Hep. and bile duct epithelium [58] b	Hep. [58] a	Hep. [60] a	Hep. [59] a	-	No detection [28] a	Yes [28] a	-
**Organ weight**	⇓ [57] b	-	-	-	-	-	-	-
**Mitochondrial effects**	-	-	NA [61] a	-	-	-	-	Yes [77] a
**Inflammatory processes**	-	-	-	-	⇑ IL-12p70 and TNF-α [37] a	-	-	-
**UPS and autophagy system**	-	-	-	-	-	Impaired [28] a	Impaired [28] a	-
**Metabolic homeostasis**	⇓ the urea cycle ⇑ blood citrulline levels ⇓ gluconeogenesis ⇓ lactate clearance [75,78] b	⇓ the urea cycle ⇑ blood citrulline levels [75] a, b	-	-	-	-	-	-

Metabolic processes mainly associated with the liver, such as gluconeogenesis and lactate clearance, were affected in the R6/2 mouse model, with a reduced hepatic glucose output following a lactate challenge, as well as alteration to several gluconeogenic enzyme activities in the mice’s livers, indicating a reduced capacity for gluconeogenesis in HD [78] (Table 3).

Bioenergetic defects were considered to be major contributing factors to neuronal dysfunction in HD [79], and in the early stages of the disease, there was an impaired glucose metabolism [80]. In post-mortem neural tissues from HD patients, there was decreased respiratory activity of mitochondria [81] due to defects in mitochondrial Complexes II, III and IV [82]. Additionally, there was an effect on the CNS during HD; examining the effect of mHTT on oxidative metabolism in liver mitochondria from 2- and 10-month-old YAC128 and WT mice did not yield evidence of mHTT-induced impairment of oxidative metabolism in YAC128 mice [61]. Mitochondrial function effects in the liver have been reported in NLS-N171-82Q H and CHL2 knock-in mouse models of HD. mHTT causes the impairment of mitochondrial function via several different mechanisms. mHTT may interact directly with mitochondria or alter transcription factors such as PGC-1a in NLS-N171-82Q HD mice [77]. mHTT directly induced mitochondrial permeability transition (MPT) pore opening in isolated mouse liver mitochondria of a CHL2 HD model [83] (Table 3).

## 7. The Retinal System

The retina is considered an extension of the brain and shares a common embryological origin with the CNS; certainly, the retina could be defined as a window to the brain. Regarding this definition, the retina allows for the study of degenerative processes from a more accessible position. However, regarding HD, the retina has not been analyzed as deeply as other areas of the CNS. 

### 7.1. Retinal Histological Characterization

The classical R6/2 model exhibits an accelerated form of the disease, and it has been reported to show reductions in rod and cone function in addition to alterations in downstream neuronal function at earlier ages (approx. 10 weeks). However, the R6/1 model, a moderate form of the disease, showed a similar pattern at a late stage of the disease (32 weeks of age) [18]. HdhQ150 mice had a similar widespread brain and peripheral pathology [84] to R6/2 mice (Table 4). 

HdhQ150 mice at 12 weeks old did not show gross morphological changes in retinal thickness, as occurred in R6/2 mouse retinas with photoreceptor layer reduction at the same time point [18,85]. The significant retinal thinning reported in the R6/2 HD model, which showed an accelerated disease progression [86], was also detected in a Drosophila model of HD [87]. The retinal thinning in R6/2 mice was related to the increased number of TUNEL-positive cells, mainly detected in the outer nuclear layer (ONL) [88]. However, the R6/1 mouse retinas showed no evidence of TUNEL-positive cells in the ONL, and there was no evidence of retinal thinning in the late stage of HD (32-week-old), as has been reported in post-mortem human retina samples [18,88].

A study of Q175 mice showed a different range of morphological abnormalities in the retina due to the cilium and retinal ciliopathy effects in HD mice [89]. Mutations in junctophilin 3 are responsible for neurodegenerative Huntington’s disease-like-2 (HDL2), a rare autosomal-dominant neurodegenerative disorder that is clinically almost indistinguishable from HD [90]. The alteration of junctophilin in the retina revealed a mixture of neurodevelopmental defects and the degeneration of the retinal neurons in Drosophila and mouse models [90,91,92]. 

Generally, Müller cell gliosis occurs when the retina is placed under stress. R6/1 mouse retinas showed increased Müller cell gliosis and the beginning of photoreceptor degeneration by 13 weeks. Reactive gliosis was related to stress events, and visual function failure due to cone-specific dysregulation and decreased cone function could contribute to increasing retinal stress [93]. A study of reactive gliosis response in HdhQ150 and R6/2 mice revealed no signs of glial cell activation, probably due to the absence of mHTT intranuclear inclusion bodies in retinal Müller cells and astrocytes, which were not affected by the deposits [85,88].

### 7.2. Retinal Nuclear Inclusion

The aggregates are accumulated, in a progressive way, as a cellular inclusion. In retinas from R6/1 and R6/2 mice, the inclusions presented a nuclear localization throughout the three neuronal layers, showing a strong prevalence when the neurological phenotype was worse. The histological and functional examination of R6/1 and R6/2 mouse retinas revealed severe retinopathy [18]. The analysis of retinas from HdhQ150 mice revealed that those nuclear inclusions were found in all neuronal cell types of the retina [85]. Interestingly, other non-neuronal retinal cells did not contain aggregates (Table 4).

### 7.3. Visual Function Evaluation

Electroretinogram (ERG) recordings evaluate visual function, and they allow for the detection of any alterations associated with HD progression. In R6/1 and R6/2 mouse retinas, the photoreceptor cell layer displayed disorganization, an irregular shape and the overall impairment of retinal response associated with strong neuronal dysfunction of both photoreceptors and retinal neurons. ERG recordings indicated that cones were affected earlier than rods and that cone- and rod-induced responses were severely reduced [93]. The R6/2 mice’s deficits in visual capabilities started as early as 4 weeks; at 8 weeks, there was severe impairment [94] (Table 4).

### 7.4. Optical Coherence Tomography

The retina is the only portion of the CNS that is optically accessible for high-resolution imaging, and it has demonstrated associations between retinal thickness and neurodegenerative disease progression. Retinal imaging technology is rapidly advancing and can provide ever-increasing amounts of information about the structure, function and molecular composition of retinal tissue in vivo. Most importantly, this information can be obtained rapidly and non-invasively through optical coherence tomography (OCT), which provides a retinal fundus image with high spatial resolution, low-cost and wide availability. Indirect ophthalmoscopy in R6/1 and R6/2 mice showed the presence of white spots all across the retina, with a progressive increase during HD [18] (Table 4).

**Table 4 ijms-23-06089-t004:** Retinal phenotype. ⇓ thinning, reduced layer thinning. NA, no alterations. ⇑ Yes, presence of described observations. ⇑ gliosis, increased gliosis. ⇓ melanopsin, reduction in melanopsin levels. ⇑ aggregates, presence of mHTT aggregates. -, no data. a, late stage. b, early stage.

Retina	Observations	Drosophila	R6/1 Mice	R6/2 Mice	Hdhq150 Mice	N171-82Q Mice
Location						
**Photoreceptors**	Thinning alterations	⇓ thinning a [87]	⇓ thinning a [18]	⇓ thinning b [18]	NA b [95]	-
	Disorganization, irregular shape and impairment	-	⇑ Yes a [93]	⇑ Yes b [94]	-	-
**ONL**	Thinning alterations	-	NA b [88]	⇓ thinning a [88]	-	-
**Müller cells**	Presence of gliosis	-	⇑ gliosis a [93]	NA a [88]	NA a [88]	-
**ipRGCs**	Changes in melanopsin concentrations	-	-	⇓ melanopsin b [96]	-	⇓ melanopsin b [97]
**Neuronal layers**	Nuclear mHTT aggregates		⇑ aggregates a [18]	⇑ aggregates a [18]	⇑ aggregates a [85]	-
**Whole retina**	Presence of white spots	-	⇑ Yes a [18]	⇑ Yes b [18]	-	-

### 7.5. Retinal Signaling Pathways Altered in HD Experimental Models

Studies of the signaling pathways involved in the neurodegenerative processes of the retina in the progress of HD are different. Many studies have focused on the potential regulatory role of chaperones and their link to the activation of intracellular misfolded protein degradation processes with the activation of proteasomes in the R6/2 mouse model and reticulum stress in the N163-Q117 mutant fly model [98,99]. Signaling studies have focused on the role of proteins related to visual signal transmission, specifically in the layers of the retina identified as responsible for visual failure in R6/2 mice, with analysis of connexin and related transcription factors [86]. 

During Huntington’s disease, circadian disruption is the most common non-motor symptom. Light/dark cycles are detected by a specific type of retinal ganglion cells called intrinsically photosensitive retinal ganglion cells (ipRGCs). Furthermore, ipRGCs respond to direct circadian photo-stimulus due to the expression of the photopigment melanopsin [100,101], and they act as an integrative station for rod/cone signals [102]. R6/2 and N171-82Q HD mouse models showed an early reduction in melanopsin levels and ipRGCs [97] associated with the histological changes detected in the neuronal number and composition of the retina (Table 4). Changes in circadian behavior in HD could be mediated by the altered molecular oscillation of two clock genes, *Period1* and *Period2*. In R6/2 mice, the *Period2* and *Period1* genes were adequately expressed during the light/dark cycle; however, the expression of the *Period2* gene was low [103,104,105].

## 8. Pancreas

### 8.1. Blood Glucose Levels

While most patients with diabetes have Type 1 diabetes (T1D) or Type 2 diabetes (T2D), there are other etiologies of diabetes associated with neurodegenerative diseases that occur less frequently. It is well described that HD patients have abnormal glucose tolerance tests and a higher prevalence of diabetes [8,106].

**Table 5 ijms-23-06089-t005:** Pancreas phenotype. ⇑ Augmented. -, no data, a, late stage, b, early stage.

Pancreas	R6/1 Mice	R6/2 Mice	N171-82Q Mice	YAC128 Mice
Observations				
**Inclusion-positive cells**	⇑ [107] a, b	⇑ [15,108] a ⇑ [109] b	⇑ [108]	-
**Type 2 diabetes mellitus-like phenotype**	⇑ [107] a, b	⇑ [108,110] a, b	⇑ [111] a, b	⇑ [112] a

Longitudinal studies of HD subjects have shown reductions in glucose utilization before the disease’s clinical onset, implying altered energy metabolism as an important component of HD pathogenesis and evidence of abnormal energy metabolism in the CNS of HD patients [113]. However, the etiology of this diabetes is unknown. HD transgenic mice develop hyperglycemia [110,114], providing an excellent experimental model for studies of diabetes mellitus and HD. It has been observed that some HD animal subjects suffer from abnormalities in energy homeostasis, and some of them develop T2D, [15,107,114], suggesting that pancreatic function is affected by mHTT (Table 5). Despite these observations, some evidence argues against the development of diabetes in HD mouse models; one report found no signs of diabetes in R6/2 mice at 8–9 weeks of age [109], and another described elevated levels of blood glucose and urine glucose in only a quarter of the 13-week-old R6/2 mice investigated [115]. N171-82Q mice demonstrated a profound dysglycemic phenotype [111]. Similarly, YAC128 mice were hyperglycemic relative to their euglycemic counterparts at 3–5 months old when fed ad libitum [112]. On the other hand, an R6/1 mouse model, although not diabetic, showed several signs of impaired glucose tolerance. Additionally, glucose tolerance in young R6/1 mice (10 weeks) was lower compared to WT mice, and in older mice (38 weeks), glucose tolerance was further impaired in both R6/1 and WT animals [107] (Table 5). 

### 8.2. Expression of Huntingtin in Pancreatic Islets 

The pathology in HD mice, as in HD patients, may be associated with the formation of nuclear inclusions derived from mHTT. Nuclear inclusions putatively derived from mHTT have been demonstrated in islets of Langerhans R6/2 mice [57] at week 7, increasing dramatically mainly in beta cells by week 12, and they have occurred in the same way in N171-82Q mice [111]. However, the expression of the native protein in the islets has not been described in animal models or clonal beta cell line 832/13 cells with the detection of HTT mRNA [108]. An HdhQ150 knock-in model expressing mutant full-length mHTT showed identical aggregate pathology distribution to R6/2 mice but only minor differences in aggregate abundance [58]. Studies using the R6/1 model showed that in the presence of the HD gene product, HTT was detected in both alpha and beta cells in R6/1 islets of Langerhans [107] (Table 5).

In an in vitro approach, the expression of N-terminal HTT fragments with different polyglutamine lengths using an insulinoma cell line (INS-1E) showed that the glucose-stimulated induction of insulin release was significantly reduced when the insulinoma cell line expressed highly expanded HTT. These results indicate that insulin release from beta cells expressing mHTT appears to be polyglutamine-length-dependent [116].

## 9. Heart

HD patients suffer from heart disease, which is the second leading cause of death in such patients [117]. These findings have also been observed in mouse models of HD, which, through electrocardiogram analysis, showed unstable heartbeats and arrhythmia in R6/1 [118], BACHD [119] and R6/2 mice [120]. This dysregulation was coupled with reduced heart size early in disease progression in R6/1 mice, and likewise in BACHD mice, significantly higher blood pressure was found compared to controls [105]. Similar results were found in pre-symptomatic HdhQ150 knock-in mice. In HdhQ150 knock-in mice, it was found that all major cardiac volumetric parameters of the heart were reduced compared to WT mice at 8 months of age, while in R6/2 mice, this was detected at 4 months of age. Although the heart mass increased in late-stage HhQ150 mice, this was not true for R6/2 mice, which could be explained by self-limiting mHTT effects or the activation of compensatory hypertrophic pathways [121]. Some studies showed that a pro-inflammatory environment, such as high levels of IL-6, is involved in cardiac remodeling by inducing left ventricular hypertrophy and increasing collagen deposits [122] and, therefore, could contribute to the cardiac pathology in BACHD mice (Table 6).

A series of changes in the heart at the cellular and tissue levels has been found in R6/2, HdhQ150 [121] and BACHD [119] animal models due to gap junctional remodeling and cardiac fibrosis, presumably as a result of the progressive loss of cardiomyocytes through apoptosis.

**Table 6 ijms-23-06089-t006:** Heart phenotype. ⇑ Augmented. -, no data. a, late stage. b, early stage.

Heart	R6/1 Mice	R6/2 Mice	BACHD Mice	HdhQ150 Knock-In
Observations				
**Inclusion-positive cells**	-	⇑ [121] a, b	-	⇑ [121] a
**Body weight and lifespan**		⇑ [115] b		-
**Arrhythmia, plasma level of noradrenaline**	⇑ [118] a, b	-	-	-
**Mitochondrial effects**	-	⇑ [121] a	⇑ [119] a, b	⇑ [121] a
**Changes in blood pressure, heart weight, contractility.**	⇑ [118] a, b	-	⇑ [119] b	-
**Cardiomyocytes failure** **(Middle to late stage)**	-	⇑ [121] a	-	⇑ [121] a

## 10. Vascular System Pericytes in Blood–Brain Barrier (BBB)

The BBB is a highly selective, semi-permeable barrier of endothelial cells that separates the blood from the brain, filtering non-essential substances from the blood into the brain and vice versa. A clinical study suggested that there is barrier leakage in HD patients and that this is possibly due to an increase in smaller blood vessels and a decrease in tight junction components [123]. Similar results were also found in an R6/2 mouse model. In this murine model of HD, tight junctions were found to be perturbed due to lower mRNA and protein levels of tight-junction-related proteins, resulting in an impaired BBB [124]. Taken together, the studies conducted indicate that pathogenic levels of HTT disrupt the filtering functions of the BBB to the extent that defects in BBB patency have been increasingly recognized not only as a component of HD but also as a therapeutic opportunity for HD treatment. Specifically, increased BBB permeability may facilitate the delivery of neurotrophic cytokines to the brain and thus promote neuroprotection and neuronal cell proliferation.

## 11. Conclusions

Taken together, the effects of HD in peripheral tissues are specific to the mouse strain and the severity the disease. Current studies regarding pathological phenotypes in peripheral tissues of HD models show specific and different effects related to the strain, experimental model and age of study. The identification of new biomarkers and mechanisms related to the progression of the disease could yield new diagnostic tools or therapeutic targets.

Since a direct toxic effect promoted by mHTT induces an inflammatory response, both the innate and adaptive immune systems could play an important role in HD, including circulation and tissue-based immune cells. An analysis of the peripheral organs should also be carried out in order to reveal the impact of inflammation on peripheral tissue homeostasis and clinical outcomes in HD. Changes in the circulating levels of cytokines are closely correlated with HD progression, and mHTT is detected in peripheral tissues as inclusions or aggregates that contribute to the perturbation of cell signaling, with pathologic implications. The effect of mHTT in peripheral tissues or organs is specific to each experimental model analyzed, and alterations are associated with HD progression status.

## Data Availability

Not applicable.

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
