# Peer review of "A New Perspective on Huntington’s Disease: How a Neurological Disorder Influences the Peripheral Tissues"

_ijms, 2022, doi:10.3390/ijms23116089_

Round 1
Reviewer 1 Report
This is a descriptive summary on the current knowledge of peripheral tissue affectation in HD animal and cell experimental models and aims to describe new therapeutic or early diagnostic options. The authors describe findings in serum/plasma, innate/adaptive immune cell subsets, spleen, pancreas, heart, retina, liver, kidney, and pericytes/BBB and describe several changes in different mouse models of HD different age/disease related status. The authors speculate that an "understanding of the impact of peripheral organ inflammation in HD may open new avenues for the development of novel therapeutic targets".
I would suggest that the authors more strictly superimpose a novel level of organization within their review, namely that they sub differentiate into N-terminal "truncated" mouse models (ie R6/2 R6/1), full length transgenic models (YAC/BAC) and knock-in models (zQ175DNki) and furthermore differentiate in early "pre-symptomatic" vs symptomatic stage.
Where ever possible that should compare more strictly deduce from human findings down to animal models and vice versa.
Type setting (e.g. references), spelling and some english corrections is advised.
Author Response
This is a descriptive summary on the current knowledge of peripheral tissue affectation in HD animal and cell experimental models and aims to describe new therapeutic or early diagnostic options. The authors describe findings in serum/plasma, innate/adaptive immune cell subsets, spleen, pancreas, heart, retina, liver, kidney, and pericytes/BBB and describe several changes in different mouse models of HD different age/disease related status. The authors speculate that an "understanding of the impact of peripheral organ inflammation in HD may open new avenues for the development of novel therapeutic targets".
I would suggest that the authors more strictly superimpose a novel level of organization within their review, namely that they sub differentiate into N-terminal "truncated" mouse models (ie R6/2 R6/1), full length transgenic models (YAC/BAC) and knock-in models (zQ175DNki) and furthermore differentiate in early "pre-symptomatic" vs symptomatic stage.
Thank you very much for your suggestions. As you have recommended we have included a specific paragraph with the different mouse models cited along the article. The review has been focused on the different experimental models including fly, cell culture and others. Certainly, the mouse model is mainly the experimental model used in experimental approaches, and we have included its sub differentiated classification. Following the reviewer indications we have included in the tables the specific early or late stage classification of the events detailed.
Where ever possible that should compare more strictly deduce from human findings down to animal models and vice versa.
Thanks a lot for your recommendation. The review has been defined in the analysis of different experimental approaches availed, and study the similarities and different between them. Recently, it has been published different great review where the authors compare the human status and compare to some animal models (Chuang, C.L et al, 2021; Przybyl, L. et al, 2021; Walaa Fakih et al, 2022).
However, throughout the text we have included some human finding down animal models, as you can find in:
-Page2 “The discovery that normal and mHTT are expressed in many tissues and organs in humans and other mammals induces the hypothesis that HD could be studied as a neural disease with a potent and unknown systemic effect, not only a condition related to Central Nervous System (CNS) dysfunctions [22].”
-Page 5: “In HD patients, IL4 levels were higher compared to healthy subjects. IL-4 is related to promoting M2 in macrophage cells or Th2 in T helper cells [14]
-Page 10: “However, the R6/1 retina showed no evidence of TUNEL-positive cells in ONL, and there was no evidence of retinal thinning in the late stage of HD (32-week-old), as has been reported in post-mortem retina human samples [18,81] “
-Page 14: ¨While most patients with diabetes have Type 1 diabetes (T1D) or Type 2 diabetes (T2D), there are other etiologies of diabetes associated with neurodegenerative diseases that occur less frequently. It is well described that HD patients have abnormal glucose tolerance tests and a higher prevalence of diabetes [8,99]”
-Page 16: “The pathology in HD mice, as in HD patients, may be associated with the formation of nuclear inclusions derived from mHTT”
-Page 17: ¨A clinical study suggested that there is barrier leakage in HD patients and that this is possibly due to an increase in smaller blood vessels and a decrease in tight junction components [117]. Similar results were also found in an R6/2 mouse model¨.
Type setting (e.g. references), spelling and some english corrections is advised.
Thank you very much for your revision. The article has been revised by English edition of MDPI service. We have attached the certificate.

Reviewer 2 Report
This interesting review paper describes recent advances in studies of peripheral effects of mutant huntingtin, a protein responsible for Huntington’s disease (HD). Although HD is a brain disorder, mutant huntingtin also affects peripheral tissues. While there are quite a bit of reviews dealing with CNS effects of mutant huntingtin, there is not many reviews summarizing the effects of mutant huntingtin on peripheral tissues. Consequently, this review paper certainly deserves attention and undoubtedly will be of interest for investigators studying HD.
For the most part, the manuscript is pretty well written; however, there are some areas of minor concern.
Page 3: “In HD patients, gamma-amino butyric acid (GABA)-releasing medium spiny neurons that are primarily expressed in the D2 dopamine receptor demonstrate an enhanced…” - “…medium spiny neurons that are primarily expressed in the D2 dopamine receptor…”, awkward, maybe the other way around: “…the D2 dopamine receptor that are primarily expressed in medium spiny neurons…”
Page 3: “In order to determine a precise diagnostic or predictive biomarker, patients' peripheral cells, therefore, provide invaluable tools and models for studying the molecular mechanisms through which endogenous HTT affects to correct physiology of peripheral tissues.” - Confusing. It is not clear what does HTT affect and how does it correct physiology of peripheral tissues.
Page 3 and throughout the manuscript: “…affectation…” – I am not sure whether the authors correctly use this word; I would replace it with “effect”.
Page 6: “The liver in a HdhQ150 knock-in mouse model (150Q) did not reach statistical significance for liver [50].” - Awkward; please, revise.
Page 8: “The recent articles revised show a reduced number of animal models.” - Confusing; I am not sure what the authors what to say here.
Page 12: “…the current studies regarding pathological phenotypes in peripheral tissues of HD models show us a specific and different affectation related to the strain, experimental model and age of study.” - Affectation ??? Age of study??? Very confusing. Please, revise.
Author Response
This interesting review paper describes recent advances in studies of peripheral effects of mutant huntingtin, a protein responsible for Huntington’s disease (HD). Although HD is a brain disorder, mutant huntingtin also affects peripheral tissues. While there are quite a bit of reviews dealing with CNS effects of mutant huntingtin, there is not many reviews summarizing the effects of mutant huntingtin on peripheral tissues. Consequently, this review paper certainly deserves attention and undoubtedly will be of interest for investigators studying HD.
For the most part, the manuscript is pretty well written; however, there are some areas of minor concern.
Page 3: “In HD patients, gamma-amino butyric acid (GABA)-releasing medium spiny neurons that are primarily expressed in the D2 dopamine receptor demonstrate an enhanced…” - “…medium spiny neurons that are primarily expressed in the D2 dopamine receptor…”, awkward, maybe the other way around: “…the D2 dopamine receptor that are primarily expressed in medium spiny neurons…”
Thank you very much for your comments. The suggestion sentence has been modified in the text.
Page 3: “In order to determine a precise diagnostic or predictive biomarker, patients' peripheral cells, therefore, provide invaluable tools and models for studying the molecular mechanisms through which endogenous HTT affects to correct physiology of peripheral tissues.” - Confusing. It is not clear what does HTT affect and how does it correct physiology of peripheral tissues.
Thank you for your appreciated commentary. The sentence has been changed in the text.
Page 3 and throughout the manuscript: “…affectation…” – I am not sure whether the authors correctly use this word; I would replace it with “effect”.
Thank you very much. We sorry the mistake. The wrong words have been changed throughout the manuscript.
Page 6: “The liver in a HdhQ150 knock-in mouse model (150Q) did not reach statistical significance for liver [50].” - Awkward; please, revise.
Thank you for the correction. The sentence has been modified in the text.
Page 8: “The recent articles revised show a reduced number of animal models.” - Confusing; I am not sure what the authors what to say here.
Sorry for the awkward written. The sentence has been revised and the text modified.
Page 12: “…the current studies regarding pathological phenotypes in peripheral tissues of HD models show us a specific and different affectation related to the strain, experimental model and age of study.” - Affectation ??? Age of study??? Very confusing. Please, revise.
Thank you very much for your suggestions. We have re-written the sentence.

Round 2
Reviewer 1 Report
The revisions are meaningful and well organized